# The Prevalence of Concomitant Abdominal Aortic Aneurysm and Cancer

**DOI:** 10.3390/jcm10173847

**Published:** 2021-08-27

**Authors:** Hyangkyoung Kim, Sung-il Cho, Sungho Won, Youngjin Han, Tae-Won Kwon, Yong-Pil Cho, Ho Kim

**Affiliations:** 1Asan Medical Center, Department of Surgery, Division of Vascular Surgery, College of Medicine, University of Ulsan, Seoul 05505, Korea; cindycrow7456@gmail.com (H.K.); medjin00@naver.com (Y.H.); twkwon2@amc.seoul.kr (T.-W.K.); ypcho@amc.seoul.kr (Y.-P.C.); 2Department of Public Health Science, Graduate School of Public Health, Seoul National University, Seoul 08826, Korea; scho@snu.ac.kr (S.-i.-C.); won1@snu.ac.kr (S.W.)

**Keywords:** abdominal aortic aneurysm, cancer, mortality, cause of death, heart failure

## Abstract

Cancers and abdominal aortic aneurysms (AAA) cause substantial morbidity and mortality and commonly develop in old age. It has been previously reported that AAA patients have a high prevalence of cancers, which has raised the question of whether this is a simple collision, association or causation. Clinical trials or observational studies with sufficient power to prove this association between them were limited because of the relatively low frequency and slow disease process of both diseases. We aimed to determine whether there is a significant association between AAA and cancers using nationwide data. The patients aged > 50 years and diagnosed with AAA between 2002 and 2015, patients with heart failure (HF) and controls without an AAA or HF matched by age, sex and cardiovascular risk factors were enrolled from the national sample cohort from the National Health Insurance claims database of South Korea. The primary outcome was the prevalence rate of cancers in the participants with and without an AAA. The secondary outcome was cancer-related survival and cancer risk. Overall, 823 AAA patients (mean (standard deviation) age, 71.8 (9.4) years; 552 (67.1%) men) and matching 823 HF patients and 823 controls were identified. The prevalence of cancers was 45.2% (372/823), 41.7% (343/823) and 35.7% (294/823) in the AAA, HF and control groups, respectively; it was significantly higher in the AAA group than in the control group (*p* < 0.001). The risk of developing cancer was higher in the AAA patients than in the controls (adjusted odds ratio (OR), 1.52 (95% confidence interval [CI], 1.24–1.86), *p* < 0.001) and in the HF patients (adjusted OR, 1.37 (1.24–1.86), *p* = 0.006). The cancer-related death rate was 2.64 times higher (95% CI, 2.22–3.13; *p* < 0.001) for the AAA patients and 1.63 times higher (95% CI, 1.37–1.92; *p* < 0.001) for the HF patients than for the controls. The most common causes of death in the AAA patients were cancer and cardiovascular disease. There was a significantly increased risk of cancer in the AAA than in the HF and control groups. Therefore, appropriate screening algorithms might be necessary for earlier detection of both diseases to improve long-term survival.

## 1. Introduction

An abdominal aortic aneurysm (AAA) is characterized by a chronic inflammatory component with a degenerative component. Its origin is a multifactorial disease related to both genetic and environmental risk factors [1]. Smoking is considered one of the most important risk factors for AAA [2]. Age is another important risk factor. While the prevalence of AAA is negligible before the age of 65 years, it increases steadily with age thereafter, estimated to range between 1% and 2% in men and around 0.5% in women aged ≥ 65 years [3]. AAAs cause 1.3% of all deaths among men aged 65–85 years in developed countries [3].

In an aging population, multimorbidity is frequent. Various diseases show an age-related prevalence, and some morbidities often coincide due to shared risk factors [4]. Cancer is not uncommon in patients with an AAA; as a result, concomitant cancer and AAA has been a long-standing subject of study in deciding the treatment priority and method [5,6]. It was reported that the annual incidence of newly diagnosed AAA in patients with cancer was 100 times higher than that in a similar age group of men in the general population, which was 0.4 to 0.67% [7]. The true incidence is difficult to determine accurately, but it has been reported to be between 0.49% and 38.1% [6,8]. Commonly reported types of associated cancers include lung, colorectal and prostate cancers [6,7,9]. Concurrence may be attributable to similar patient demographics such as advanced age and male sex and common risk factors such as smoking [10]. There may also be a common pathway, as suggested in other diseases [11,12,13]. Traditionally, AAA is regarded as a consequence of atherosclerosis owing to its association with the atherosclerotic change of the aortic wall [14]. However, recent studies have demonstrated that medial and adventitial injuries from proteolysis, oxidative stress and adaptive immune responses, rather than atherosclerotic change, are involved [15,16,17]. Oxidative stress and its direct consequences, including lipid peroxidation, play a role in the malignancy and were suggested as a linkage between cancers and AAAs [18].

There have been sporadic reports on the coexistence of the two diseases so far, but there have been no large-scale studies involving a sufficient number of subjects to date on the exact prevalence of the two diseases. In this study, we aimed to investigate the prevalence of concomitant AAA and cancer in a national sample cohort from the South Korean population. In addition, the mortality rate related to AAA and the cause of death were analyzed.

## 2. Methods

### 2.1. Data Sources

This study was undertaken using the national sample cohort (NSC) database accumulated from January 2002 to December 2017 which was acquired from the National Health Insurance Service (NHIS). The NHIS-NSC database is a population-based cohort established by the NHIS. This cohort includes approximately 2.2% of the entire eligible population randomly sampled from the 2002 National Health Insurance Recipient Qualifications Database and followed up until 2015. Each patient’s demographic information, International Classification of Disease, Tenth Revision (ICD-10) diagnosis codes, procedure codes and survival in inpatient and outpatient services were collected and analyzed. Approval for data collection and publication was granted by the institutional review board (IRB No. 2020-0643) of our hospital, which waived the requirement for written informed consent because of the retrospective nature of the study and the lack of information on the participant’s identification. All the methods were performed in accordance with the relevant guidelines and regulations.

### 2.2. Study Design and Cohort Definition

First, the patients diagnosed with AAA (ICD-10 codes I71.3-4 and I71.8-71.9) between January 2002 and December 2015 were screened. The following patients were then excluded: those with ICD-10 code I71.9, diagnosed with AAA before June 2002 or after June 2015 and/or aged younger than 50 years. Finally, patients with the following conditions were excluded: (1) AAA related to Behcet’s disease (ICD-10 code M35.2) or syphilis (ICD-10 code A53.9), (2) history of typhoid fever (ICD-10 code A01) and/or (3) age of AAA diagnosis younger than 50 years. The two control groups without AAA were randomly sampled from individuals who had not been diagnosed with AAA during the same period, one with heart failure (HF (ICD-10 codes I50.0-9)) and the other without HF after matching for age, sex and other cardiovascular risk factors such as hypertension (HTN), diabetes (DM) and dyslipidemia (1:1:1 matching). The HF group was used as the control group for comparison because the association of HF with cancer is relatively well–known. The sex was matched completely, the age difference was less than 6 years and HTN, DM and dyslipidemia were matched where applicable. The index date of each group was defined as the first diagnosis date of AAA in the AAA group or of HF in the HF group. In the control group, the index date was 1 June of the corresponding year of the matched AAA patients.

### 2.3. Study Outcomes

The primary outcome was the development of any cancer. Patients with cancer were defined as those who had one or more hospitalizations, visited outpatient clinics at least twice and/or had the Special Support for Serious Illness Act code with a principal or sub- diagnosis code of cancer (ICD-10 codes C00–97). The Special Support for Serious Illness Act code for cancer is given only when histologically proven. The cancer diagnosis time was classified as follows based on the index date: at least 6 months before the index date, within 6 months before and after the index date, and 6 months after the index date. Cancer that occurred at least 6 months after the index date was defined as a subsequent cancer. Initially, cancer risk was analyzed by including all cancers regardless of the period of occurrence. The hazard ratio was then calculated by analyzing patients with newly developed cancer among those who did not develop cancer until 6 months from the index date. The secondary outcome was the cause of death and the mortality rate per 100 person-years.

### 2.4. Statistical Analysis

Demographic data were compared using a generalized estimating equation method with appropriate distributions for paired data. The association with cancer was analyzed by conditional logistic regression and the results were represented as odds ratios (ORs) and 95% confidence intervals (CIs). The incidence of cancer or death after enrollment was evaluated with the incidence rate per 100 person-years and the Kaplan–Meier curve analysis. A Cox proportional hazards regression model with a robust variance estimate considering paired data was used to determine the adjusted hazard ratio (HR) and corresponding 95% CIs for the association between AAAs and cancers. The cumulative survival probability was presented graphically with a Kaplan–Meier curve. All *p*-values < 0.05 were considered significant. The statistical analysis was performed using SAS Enterprise Guide version 7.1 (SAS Institute Inc., Cary, NC, USA) and R version 3.0.3 (R Development Core Team, 2006).

## 3. Results

### 3.1. Verification of ICD-10 Codes

First, to accurately identify patients with an AAA, the ICD-10 code corresponding to AAA was verified using hospital data. The image sets and medical records of patients from a single university hospital database from 1 January 2018 to 31 December 2018 searched by AAA codes and codes of the diseases similar to AAA were reviewed. The AAA codes were I71.3–4 and I71.8. Codes of the diseases similar to AAA included thoracoabdominal aortic aneurysm (I71.6), dissection of aorta (I71.0), thoracoabdominal aortic aneurysm, ruptured (I71.5), aortic aneurysm of unspecified site, without rupture (I71.9), aneurysm and dissection of iliac artery (I723), aneurysm and dissection of the artery of lower extremity (I72.4) and aneurysm of the aorta in diseases classified elsewhere (I79.0). The computed tomography images and medical records were reviewed by one vascular surgeon. The vascular surgeon independently investigated whether the registered and similar codes matched the diagnosis of AAA and calculated the specificity and sensitivity. Upon reviewing the hospital data of 2352 patients, it was decided that the I71.9 code would not be included in this study since the misclassification rate was high: 35 patients out of 52 patients (67.3%) were not AAA patients. The sensitivity and specificity of I71.3–4 and I71.8 were 99.7% (642/644) and 99.6% (1650/1656), respectively.

### 3.2. Characteristics of the Population

Among the 1,108,369 subjects of the NHIS-NSC database, we identified 899 patients with AAA, 76 of whom were excluded from the analysis according to the aforementioned criteria (Figure 1). The AAA group’s mean age was 71.8 years (standard deviation (SD), 9.4 years), and the group was comprised of 32.9% (271) women and 67.1% (552) men. Among the remaining 1,106,687 participants without AAA, through 1:1:1 matching, we included 823 patients to the HF cohort and 823 patients to the control cohort. The baseline characteristics of the three groups are summarized in Table 1. The smoking and alcohol history data of 37.9% of the AAA/HF/control cohort were missing.

### 3.3. Association between AAAs and Cancers

#### 3.3.1. Prevalence and Incidence of Cancers

A total of 372 (45.2%) AAA, 343 (41.37%) HF, and 294 (35.7%) control patients were diagnosed with malignant cancer (*p* < 0.001). The period when the patient was diagnosed with cancer in relation to the index date and the type of cancer is described in Table 2. In the AAA group, the time of cancer diagnosis was as follows: more than 6 months prior to the index date in 177 (47.6%) patients, within 6 months before and after the index date in 103 (27.7%) patients and more than 6 months after in 92 (24.7%) patients.

In terms of breast and genitourinary cancer, cancer types were masked and indistinguishable in the main diagnostic codes of the NHIS-NSC database due to its privacy policy. Among the identifiable types of cancer, the most common type in the AAA group was colorectal cancer (31.2%, 116/372), followed by lung cancer (15.3%, 57/372). The most common cancer in the HF group and the control group was lung cancer, accounting for 19.5% (67/343) and 11.6% (34/294) of the cases, respectively.

#### 3.3.2. Comparison of the Risk of Cancer between the Groups

Cancer risk was analyzed by the cancer prevalence rate in each group. The risk of cancer was 1.50 times higher (95% CI, 1.23–1.83) in the AAA group and 1.29 times higher (95% CI, 1.06–1.58) in the HF group than in the control group (*p* < 0.001) (Figure 2). The risk of cancer was significantly higher in the AAA group regardless of the adjustment. When adjusted by HTN, DM, dyslipidemia and smoking, the risk of cancer was 1.47 times higher (95% CI, 1.11–1.96) in the AAA group and 1.45 times higher (95% CI, 1.02–2.07) in the HF group than in the control group (*p* = 0.007 and *p* = 0.038, respectively). When comparing the AAA and HF groups, the difference between both groups was not statistically significant (crude OR, 1.16 (95% CI, 0.95–1.41), *p* = 0.144; adjusted OR, 1.02 (95% CI, 0.73–1.44), *p* = 0.895).

#### 3.3.3. Cancer Risk by Age Groups

The age-specific cancer risk was analyzed (Table 3). In the patients with AAA who were younger than 65 years old, the cancer risk was 2.01 times that in the control group (95% CI, 1.27–3.18, *p* = 0.003). Similarly, the HF patients younger than 65 years old showed a 2.20 times higher risk than the control group (95% CI, 1.40–3.47, *p* = 0.001). In the patients 65 years of age or older, the patients with AAA had a significantly higher risk than the control group, while the HF and control groups had a similar risk (*p* = 0.004 and *p* = 0.300, respectively).

### 3.4. Risk of Subsequent Cancer

The risk of cancer in patients without a history of cancer or without cancer until after 6 months after the index date is described in Table 4. The cancer development rate per 100 person-years was 6.87 in the AAA group, 4.88 in the HF group and 3.89 in the control group. The unadjusted HRs were 1.72 (95% CI, 1.35–2.21) for the AAA group (*p* < 0.001) and 1.28 (95% CI, 1.02–1.60) for the HF group (*p* = 0.035) in comparison to that for the control group. The HRs adjusted by age, sex, DM, hypertension and dyslipidemia were 0.71 (95% CI, 0.56–0.88) for the AAA group (*p* = 0.002) and 0.56 (95% CI, 0.43–0.72) for the HF group (*p* < 0.001) in comparison to that for the control group.

In the patients without cancer until after 6 months after the index date, the cancer risk was not significantly different between the groups (*p* = 0.199) (Figure 3).

### 3.5. Comparison of the Mortality Rate

The mortality rate of each group is summarized in Table 5. The mortality rate per 100 person-years was 11.11 in the AAA group, 6.80 in the HF group and 4.40 in the controls. The patients with AAA had a 2.82 times higher mortality risk than the controls (*p* < 0.001). When compared to the HF group, the AAA patients had a higher mortality rate (*p* < 0.001). In the AAA group, the 5-year survival rate was 57.7% (95% CI, 53.7–61.6). The 5-year survival rate of the HF and control groups were 75.17% (95% CI, 72.2–78.1) and 80.87% (95% CI, 77.7–84.0), respectively.

#### Causes of Death

The causes of death are summarized in Table 6. The most common causes of death in the AAA group were cancer and ruptured AAA, followed by cardiovascular and cerebrovascular diseases. In the HF and control groups, the most common causes of death were cancer and cardiovascular and cerebrovascular diseases. The most common type of cancer related to death was lung cancer. Prostate cancer was not identifiable with the diagnosis code in the database, but it was listed in the cause of death database. It was the second most common cancer as a cause of death in the AAA and HF groups and the fourth in the controls. Cardiovascular death with AAA rupture-related death was detected in 64 (7.8%) patients in the AAA group, which was the same frequency as that of cancer-related death.

## 4. Discussion

The association between AAAs and cancers has been suggested in previous studies, but it was difficult to clarify because it was difficult to obtain a number of patients large enough to have sufficient power. This study aimed at evaluating the possible association between AAAs and cancers. In this study, we analyzed each of the 823 AAA patients, HF patients and controls. To the best of our knowledge, this study is one of the first publications with a large validated sample cohort of patients with AAA demonstrating the risk of cancer through a comparison with patients with HF, which is well-known for having a high cancer risk, as well as a control group without AAA. The NHIS-NSC database is a representative database constructed by systematic stratified random sampling with a proportional allocation within each stratum according to the individual’s total annual medical expenses [19]. Notably, the ICD-10 code for cancer is validated in its accuracy by comparison with the Korea National Cancer Incidence Database built by the Korea Central Registry, which only registers histologically proven malignancies [20]. Moreover, to maximize the accuracy of selection of patients with cancer, we defined these patients as those who visited outpatient clinics at least twice a year with the same code. The reason to select HF as one of the control groups was that there had been many previous reports suggesting an association between cancer and patients with HF [12,13,21,22]. In our study, the risk of cancer in the patients with AAA was the highest among the three groups (45.2% for the AAA group vs 41.7% for the HF group vs. 35.7% for the controls, Table 2). The AAA group especially showed a significantly higher cancer risk, which was 1.52 times higher than that in the control group. Compared to the HF group, the risk was similar (crude OR, 1.11, *p* = 0.340, Figure 2). Therefore, the findings from this study suggest that the two diseases coexist not only occasionally or are caused by the old age of patients, but are also strongly associated.

Cardiovascular diseases are a significant cause of death for many cancer survivors and rivals cancer recurrence [23]. Conversely, cardiovascular diseases are associated with a higher incidence of cancer [24,25]. Previous registry-based cohorts of patients with myocardial infarction showed a modest 5–8% increased risk of cancer [26,27,28]. In addition, the incidence of cancer was increased among patients with preexisting HF, with the estimated incidence in the range of 18.9–33.7 per 1000 person-years [24,29,30,31]. Similarly to other cardiovascular diseases, AAA often coexists with cancer. In accordance with previous reports, our data revealed a high prevalence of cancer in patients with AAA, accounting for 45.2% of these patients (372/823). The suggested mechanisms of coexistence of AAA and malignancies include common risk factors, proinflammatory conditions and oxidative stress. AAA shares a number of modifiable risk factors with cancer, such as smoking and increased age [4,8,32]. It has also been suggested that it might be due to the presence of chronic inflammatory cells and cytokines and the angiogenesis status in patients with AAA [32]. Another possible mechanism is the disturbed interaction between matrix proteins and epithelial cells, which facilitates angiogenesis or carcinoma and the development of an aneurysm-prone phenotype [33,34]. Surveillance during the active follow-up of AAA or cancer may also result in a growing prevalence of coexistence [24,35]. Similar findings were observed in the experimental study, and systemic pathological processes, such as inflammation and oxidative stress, are among the main hypotheses, possibly superimposed on the background of genetic predisposition [36]. Circulating neurohormonal factors were also shown to affect tumor biology [21].

When stratified by age group in our study, the increased cancer risk was noticeable in patients under the age of 65 years in both the AAA and HF groups (Table 3). In the AAA group, an increase in cancer risk was observed regardless of age but was more pronounced in patients under 65 years of age. Interestingly, in the HF group, an increase in cancer risk was no longer observed in patients over 65 years of age. Regarding the temporal sequence of AAA and cancer, the cancer diagnosis appeared earlier than the AAA diagnosis (75.3% (280/372)). When comparing each group, the AAA group had a higher cancer risk before the index date whereas the HF group had an increased subsequent cancer risk (Table 2). However, since both diseases progress slowly and the order of discovery does not indicate the order of onset, it is difficult to conclude which disease precedes the other.

In our study, the mortality rate was 2.64 times higher in the AAA group than in the control group (*p* < 0.001, Table 5). However, when adjusted by age, sex and underlying condition, the mortality risk was lower than in the control group (adjusted HR, 0.51 (95% CI, 0.41–0.65)). These findings suggested that controlling associated risk factors along with the management of AAA may contribute to an increased survival rate. In our study, the most common cause of death in the patients with AAA was cancer, followed by ruptured AAA. As indicated in our study, cancer was associated with a significantly high prevalence and was a major cause of death in the AAA group. A substantial mortality risk was also associated with ruptured AAA, although it was shown in previous reports to decline gradually over time [37,38] Therefore, it seems necessary to run efficient and cost-effective screening programs for these diseases in those with the highest risk.

This study had several limitations. Firstly, other than the high rate of coexistence of AAA and cancer, the association between the two diseases including common risk factors and shared causation could not be revealed due to the retrospective observational nature of this study. Secondly, the national database may have included some misclassification of the diagnosis. To get the most accurate target patients possible, however, we pre-evaluated hospital data and included only accurate diagnostic codes to our analysis. In addition, we excluded the AAAs related to distinct mechanisms such as Behcet’s disease from the study. Thirdly, this claim data offered limited clinical data, including smoking history, because such information was included only for the patients who had undergone a national health check-up. As a result, it was not possible to hypothesize some potential mechanisms of the association between AAAs and cancers. We attempted to analyze only patients with such information, but it seemed more likely that the data loss was too large to represent the population. Instead, we analyzed the risk ratio by adjusting these risk factors for people with such information, which also showed a significant difference. Fourthly, we tried to eliminate infective AAA, but diseases related to typhoid fever and syphilis were masked from the database via categorization into sensitive diagnoses with regard to the patients’ privacy. As the number of patients with Behcet’s disease was as small as 11, we assumed it to be a negligible number. Finally, there was no information whether the control group had no undiagnosed cancer or AAA, and there was a possibility that the higher rate of coexistence might have resulted from increased detection of other diseases during workup for one disease. The finding that the cancer detection rate was high within 6 months of the AAA diagnosis supports this possibility. However, we believe that this did not solely come from the imaging workup because the cancer detection rate 6 months after the index date in the control group was similar to that in the AAA group. In addition, subsequent cancer incidence per 100 person-years and cancer-related mortality was higher in the AAA group than in the control group. Despite these limitations, the current study had its strengths. First, it is the largest study to date that has investigated the association in patients with an AAA and cancer, considering that it has been relatively infrequently reported that both AAA and cancer coexist in one patient. It seems important to elucidate the prevalence of coexistence to suggest simultaneous surveillance for the improvement of long-term survival or elucidate a possible common pathogenesis or risk factors. Moreover, we matched the patients with AAA to those with HF, whose association with cancer has already been suggested, as well as with a control group, supporting our findings. We believe our study can stimulate further research to delineate a connection between AAAs and cancers, shed light on possible mechanisms and risk factors and eventually develop cost-effective and practical surveillance protocols.

## 5. Conclusions

There was a significantly increased risk and higher prevalence of cancer in the AAA group than in the control group in this 13-year cohort. Further research is needed to elucidate the possible shared pathologic process. In the meantime, effective screening programs for both diseases need to be developed.

## Figures and Tables

**Figure 1 jcm-10-03847-f001:**
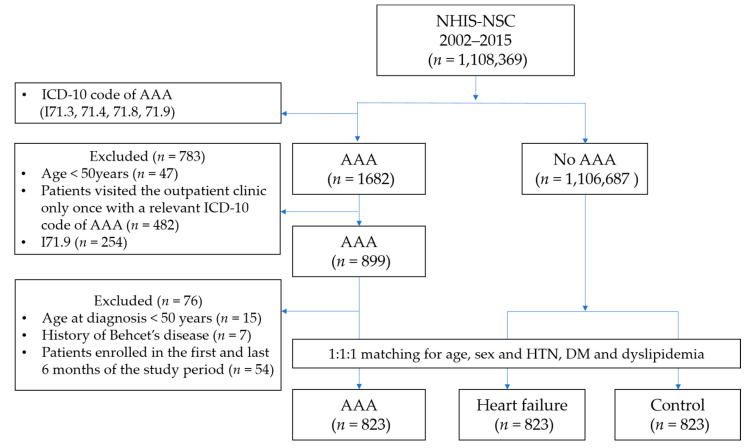
Flow diagram.

**Figure 2 jcm-10-03847-f002:**
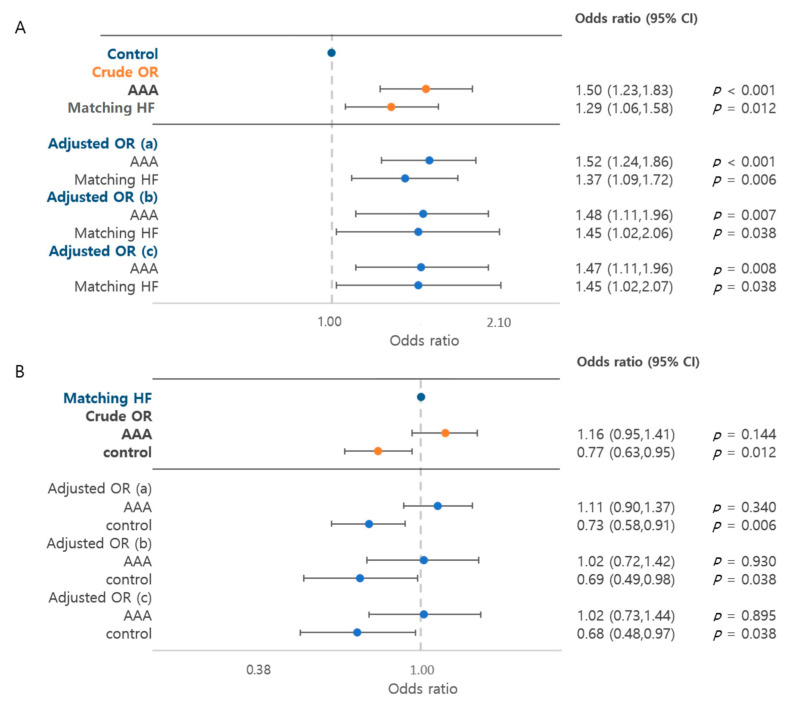
Risk of cancer. (**A**) Control group as a reference. (**B**) Matching HF group as a reference. AAA, abdominal aortic aneurysm; HF, heart failure. (a) Adjusted by hypertension, diabetes, dyslipidemia; (b) adjusted by hypertension, diabetes, dyslipidemia, smoking; (c) adjusted by hypertension, diabetes, dyslipidemia, smoking, alcohol.

**Figure 3 jcm-10-03847-f003:**
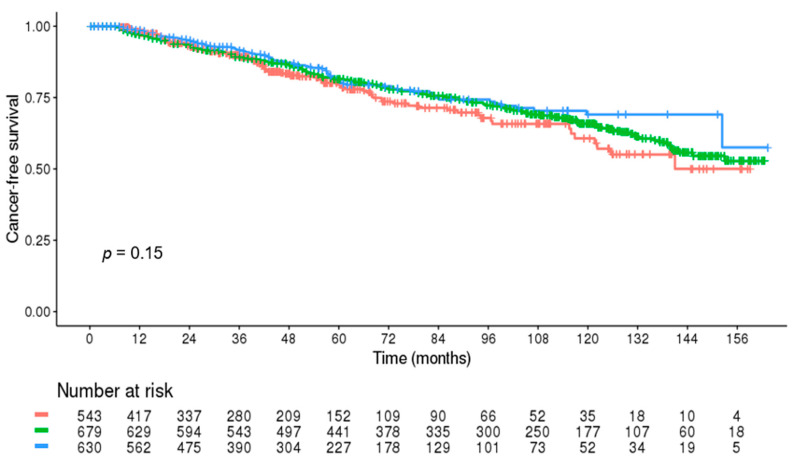
Cancer-free survival during the long-term follow-up in the patients without cancer until after 6 months after the index date (red line: abdominal aortic aneurysm group; green line: heart failure group; blue line: control group).

**Table 1 jcm-10-03847-t001:** Baseline clinical characteristics of the patients.

	AAA (*n* = 823)	Matching HF Patients (*n* = 823)	Matching Controls (*n* = 823)	*p*
Age (years), mean (SD)	71.8 (9.4)	71.6 (9.2)	71.8 (9.4)	0.903
Male (*n* (%))	552 (67.1)	552 (67.1)	552 (67.1)	>0.999
Comorbidities				
Diabetes mellitus	512 (62.2)	487 (59.2)	512 (62.2)	<0.001
Hypertension	716 (87.0)	717 (87.1)	716 (87.0)	0.607
Dyslipidemia	619 (75.2)	413 (50.2)	716 (87.0)	<0.001
Alcohol *				0.087
None	372 (45.2)	256 (31.1)	362 (44.0)	
<3 d/week	111 (13.5)	79 (9.6)	148 (18.0)	
≥3 d/week	71 (8.6)	51 (6.2)	82 (10.0)	
Missing	269 (32.7)	437 (53.1)	231 (28.1)	
Smoking *				<0.001
Non-smoker	286 (34.8)	261 (31.7)	390 (47.4)	
Ex-smoker	64 (7.8)	47 (5.7)	90 (10.9)	
Current smoker	204 (24.8)	79 (9.6)	112 (13.6)	
Missing	269 (32.7)	436 (53.0)	231 (28.1)	

* Smoking and alcohol history was obtained from the general health check-up data from the index year to the following year. Abbreviations used: AAA, abdominal aortic aneurysm; HF, heart failure; SD, standard deviation.

**Table 2 jcm-10-03847-t002:** Cancer diagnosis period in relation to the index date in each group.

	AAA (*n* = 823)	Matching HF Patients (*n* = 823)	Matching Controls (*n* = 823)
Cancer (*n* (%)) *	372 (45.2)	343 (41.7)	294 (35.7)
Cancer diagnosis period in relation to the index date (*n* (%)) *			
More than 6 months before	177 (47.6)	97 (28.3)	166 (56.5)
Within 6 months before	56 (15.1)	18 (5.2)	15 (5.1)
Up to 6 months after	47 (12.6)	29 (8.5)	12 (4.1)
More than 6 months after	92 (24.7)	199 (58.0)	101 (34.4)
Types of cancer			
Lung	57 (15.3)	67 (19.5)	34 (11.6)
Colorectal	116 (31.2)	38 (11.1)	37 (12.6)
Hematologic	12 (3.2)	7 (2.0)	2 (0.7)
Other solid tumor †	187 (50.3)	231 (67.3)	221 (75.2)

* Note: *p* < 0.001. † Breast and genitourinary cancers are included. Abbreviations used: AAA, abdominal aortic aneurysm; HF, heart failure.

**Table 3 jcm-10-03847-t003:** Cancer risk stratified by age groups (≥ 50 and < 65 years and ≥ 65 years).

	AAA	Matching HF Patients	Matching Controls
Age ≥ 50 and < 65 years	*n* = 196	*n* = 196	*n* = 196
Cancer	74 (37.8)	78 (39.8)	47 (24.0)
Crude OR (95% CI)	2.01 (1.27–3.18)	2.20 (1.40–3.47)	1 (reference)
*p*	0.003	0.001	
Adjusted ^a^ OR (95% CI)	2.02 (1.28–3.19)	2.27 (1.43–3.60)	1 (reference)
*p*	0.003	0.001	
Adjusted ^b^ OR (95% CI)	2.43 (1.33–4.42)	2.40 (1.19–4.84)	1 (reference)
*p*	0.004	0.014	
Age ≥ 65 years	*n* = 627	*n* = 627	*n* = 627
Cancer	298 (47.5)	265 (42.3)	247 (39.4)
Crude OR (95% CI)	1.40 (1.12–1.75)	1.13 (0.90–1.41)	1 (reference)
*p*	0.004	0.300	
Adjusted ^a^ OR (95% CI)	1.42 (1.13–1.78)	1.17 (0.91–1.49)	1 (reference)
*p*	0.003	0.225	
Adjusted ^b^ OR (95% CI)	1.28 (0.93–1.76)	1.28 (0.87–1.89)	1 (reference)
*p*	0.137	0.206	

^a^ Adjusted by hypertension, diabetes, dyslipidemia; ^b^ adjusted by hypertension, diabetes, dyslipidemia, smoking. Abbreviations used: AAA, abdominal aortic aneurysm; HF, heart failure; OR, odds ratio; CI, confidence interval.

**Table 4 jcm-10-03847-t004:** Cancer incidence in the subjects without cancer until after 6 months after the index date.

	AAA (*n* = 823)	Matching HF Patients (*n* = 823)	Matching Controls (*n* = 823)
Number of patients without cancer until after 6 months after the index date	590	708	642
Number of those with subsequent cancer	92	199	101
Cancer incidence per 100 person-years	4.36	4.16	3.43
Unadjusted HR (95% CI)	1.30 (0.97–1.72)	1.14 (0.90–1.46)	1 (reference)
*p*	0.075	0.285	
	1.15 (0.90–1.47)	1 (reference)	0.87 (0.68–1.10)
*p*	0.276		0.245
Adjusted ^a^ HR (95% CI)	1.39 (1.04–1.47)	1.16 (0.89–1.51)	1 (reference)
*p*	0.025	0.273	
	1.20 (0.93–1.55)	1 (reference)	0.86 (0.66–1.12)
*p*	0.170		0.273
Adjusted ^b^ HR (95% CI)	1.48 (1.05–2.08)	1.23 (0.87–1.74)	1 (reference)
*p*	0.027	0.244	
	1.20 (0.85–1.69)	1 (reference)	0.81 (0.57–1.15)
*p*	0.295		0.244
Adjusted ^c^ HR (95% CI)	1.51 (1.07–2.13)	1.26 (0.89–1.79)	1 (reference)
*p*	0.020	0.192	
	1.19 (0.85–1.68)	1 (reference)	0.79 (0.56–1.12)
*p*	0.308		0.192

^a^ Adjusted by age, sex, DM, HTN and dyslipidemia; ^b^ adjusted by age, sex, DM, HTN, dyslipidemia and smoking; ^c^ adjusted by age, sex, DM, HTN, dyslipidemia, smoking and alcohol. Abbreviations used: AAA, abdominal aortic aneurysm; HF, heart failure; HR, hazard ratio; CI, confidence interval.

**Table 5 jcm-10-03847-t005:** Comparison of the mortality rate.

	AAA (*n* = 823)	HF (*n* = 823)	Matching Controls (*n* = 823)
Number (%) of deaths	343 (41.7)	433 (52.6)	170 (20.7)
Mortality rate per 100 person-years	11.11	6.80	4.40
5-year survival rate (%)	57.66 (53.74–61.58)	75.17 (72.21–78.13)	80.87 (77.73–84.01)
Unadjusted HR (95% CI)	2.64 (2.22–3.13)	1.63 (1.37–1.92)	1 (reference)
*p*	< 0.001	< 0.001	
	1.61 (1.42–1.84)	1 (reference)	0.65 (0.55–0.76)
*p*	< 0.001		< 0.001
Adjusted ^a^ HR (95% CI)	2.82 (2.34–3.39)	1.52 (1.25–1.84)	1 (reference)
*p*	< 0.001	< 0.001	
	1.86 (1.59–2.17)	1 (reference)	0.67 (0.54–0.80)
*p*	< 0.001		< 0.001
Adjusted ^b^ HR (95% CI)	2.95 (2.30–3.77)	1.51 (1.14–2.01)	1 (reference)
*p*	< 0.001	0.005	< 0.001
	1.95 (1.56–2.45)	1 (reference)	0.66 (0.50–0.88)
	< 0.001		0.005
Adjusted ^c^ HR (95% CI)	2.89 (2.25–3.70)	1.50 (1.12–1.99)	1 (reference)
*p*	< 0.001	0.006	< 0.001
	1.93 (1.54–2.42)	1 (reference)	0.67 (0.50–0.89)
	< 0.001		0.006

^a^ Adjusted by age, sex, DM, HTN and dyslipidemia; ^b^ adjusted by age, sex, DM, HTN, dyslipidemia and smoking; ^c^ adjusted by age, sex, DM, HTN, dyslipidemia, smoking and alcohol. Abbreviations used: AAA, abdominal aortic aneurysm; HF, heart failure; HR, hazard ratio; CI, confidence interval.

**Table 6 jcm-10-03847-t006:** Major causes of death.

Category of Death	AAA	HF	Control	Total
Total number of deaths	340 (41.3)	431 (52.4)	170 (20.7)	941 (38.1)
Cancer	64 (7.8)	82 (10.0)	45 (5.5)	191 (7.7)
Lung	23 (2.8)	21 (2.6)	11 (1.3)	55 (2.2)
Stomach	4 (0.5)	7 (0.9)	6 (0.7)	17 (0.7)
Prostate	3 (0.4)	10 (1.2)	2 (0.2)	15 (0.6)
Colorectal	5 (0.6)	7 (0.9)	2 (0.2)	14 (0.6)
Hematologic	4 (0.5)	5 (0.6)	3 (0.4)	12 (0.5)
Cardiovascular	31 (3.8)	46 (5.6)	13 (1.6)	90 (3.6)
Cerebrovascular	27 (3.3)	36 (4.4)	10 (1.2)	73 (3.0)
Ruptured AAA	64 (7.8)	0	0	64 (2.6)
COPD	10 (1.2)	29 (3.5)	4 (0.5)	43 (1.7)
Pneumonia	13 (1.6)	17 (2.1)	7 (0.9)	37 (1.5)
Intracranial hemorrhage	8 (1.0)	7 (0.9)	5 (0.6)	20 (0.8)

Each cell contains *n* (%). Abbreviations used: AAA, abdominal aortic aneurysm; HF, heart failure; COPD, chronic obstructive pulmonary disease.

## Data Availability

The data presented in this paper are not available outside of the National Health Insurance System server because downloading is prohibited. They are assessable after authorization by the National Health Insurance System.

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
