# Peer review of "The Prevalence of Concomitant Abdominal Aortic Aneurysm and Cancer"

_jcm, 2021, doi:10.3390/jcm10173847_

Round 1

Reviewer 1 Report

Many thanks for asking me to review the manuscript. My main concern is the interpretation of the data, as you will see from my comments to the authors. Possible confounders should not be overlooked and, in my opinion, should be discussed further  in the manuscript and/or an associated commentary/editorial. 

Congratulations for a well written manuscript, which documents an association between a diagnosis of AAA and one of cancer. The authors’ evidence, based on registry data, is in line with previous literature, as quoted in discussion.
The authors highlight possible explanations for this association and even propose some mechanistic theories of shared causation, which remain unproven. The authors, however, overlook the fact that patients with “serious” conditions, such as AAA or heart failure, are often investigated far more thoroughly than the general population (“controls”). For example, most patients with cancer would undergo cross-sectional imaging of the abdomen for staging purposes, and would thus be effectively screened for AAA. Similarly, investigation of patients with AAA frequently reveal hidden cancers. Incidental AAAs are thus often discovered during the process of cancer staging. Equally, many solid tumours are detected on aortic CT scans, in potential candidates for aortic surgery. It should be clear that absence of a diagnosis of cancer (or AAA) does not equate to absence of cancer (or AAA). It is thus possible that, in this study, several “controls” had undiagnosed cancers (or AAAs), which would have become apparent had they undergone the same investigations of the patients in the AAA group. It is of interest that ¾ cancers were diagnosed shortly before the presence of AAA was detected, suggesting that the AAA could have been uncovered by investigations triggered by the cancer diagnosis. This possibility is also evident in table 2, when comparing AAA patients and controls: the difference in cancer diagnoses occurred almost exclusively in the 6 months before and after the diagnosis of AAA. Further, the incidence of new cancers detected six months after the diagnosis of AAA was similar to the incidence among control patients. This issue highlights the limitation of retrospective reviews of registry data, which are rarely granular enough to provide definite answers to such questions. Finally, it is worth commenting about the significant missing data on smoking habits (yet tobacco use was apparently more prevalent in the AAA group), thus the inability to adjust analyses for one of the most potent risk factors for certain cancers and AAA.

I would ask that the above issues are highlighted and appraised in the discussion.

Author Response

Response to Reviewer 1 comments

Thank you for your comments. We made the following changes:

Many thanks for asking me to review the manuscript. My main concern is the interpretation of the data, as you will see from my comments to the authors. Possible confounders should not be overlooked and, in my opinion, should be discussed further  in the manuscript and/or an associated commentary/editorial. 

Congratulations for a well written manuscript, which documents an association between a diagnosis of AAA and one of cancer. The authors’ evidence, based on registry data, is in line with previous literature, as quoted in discussion.
The authors highlight possible explanations for this association and even propose some mechanistic theories of shared causation, which remain unproven. The authors, however, overlook the fact that patients with “serious” conditions, such as AAA or heart failure, are often investigated far more thoroughly than the general population (“controls”). For example, most patients with cancer would undergo cross-sectional imaging of the abdomen for staging purposes, and would thus be effectively screened for AAA. Similarly, investigation of patients with AAA frequently reveal hidden cancers. Incidental AAAs are thus often discovered during the process of cancer staging. Equally, many solid tumours are detected on aortic CT scans, in potential candidates for aortic surgery. It should be clear that absence of a diagnosis of cancer (or AAA) does not equate to absence of cancer (or AAA). It is thus possible that, in this study, several “controls” had undiagnosed cancers (or AAAs), which would have become apparent had they undergone the same investigations of the patients in the AAA group. It is of interest that ¾ cancers were diagnosed shortly before the presence of AAA was detected, suggesting that the AAA could have been uncovered by investigations triggered by the cancer diagnosis. This possibility is also evident in table 2, when comparing AAA patients and controls: the difference in cancer diagnoses occurred almost exclusively in the 6 months before and after the diagnosis of AAA. Further, the incidence of new cancers detected six months after the diagnosis of AAA was similar to the incidence among control patients. This issue highlights the limitation of retrospective reviews of registry data, which are rarely granular enough to provide definite answers to such questions. Finally, it is worth commenting about the significant missing data on smoking habits (yet tobacco use was apparently more prevalent in the AAA group), thus the inability to adjust analyses for one of the most potent risk factors for certain cancers and AAA.

I would ask that the above issues are highlighted and appraised in the discussion.

Response: We strongly agree with the reviewer’s point. We added it to the Discussion section. There is no way to prove that the control group did not have AAA, HF, or cancer. Moreover, it is true that more cancers were discovered during the workup of AAA, which is suggested by Table 2. One strong point of our study is the long-term follow-up. This study population comprises of 13 years of cohorts. From a chronological point of view, it is reasonable to think that cancer discovered during the workup of AAA occurred long before the AAA diagnosis. If the incidence of cancer was not actually high in AAA patients, but simply that much more cancer cases were detected during the AAA workup process, the probability of cancer detected after the index date in the control group should be much higher; however, this was not the case. Subsequent cancer incidence per 100 person-years (Table 4) and cancer-related mortality (Table 6) was higher in the AAA group than in the control group. We believe that this can support the higher probability of the coexistence of cancer in AAA patients when comparing that in people of the same age group.

Reviewer 2 Report

The authors cover an interesting issue on regarding the co-prevalence of AAA and cancer and the possible implication for clinical decision making by investigating death rates in three different selected populations. This data is currently missing and should be of great interest to clinical practitioners. Although of general interest, the study harbors many major and minor problems in its current form.

Please find a list of topics that need to be addressed before eventual publication. PLease do consider an extensive revision of your manuscript, since the data would be interesting to the readership. 

General:

- The English language is hard to understand and covers a variety of syntax and grammatical errors as well as typos. Please revise with a native speaker or professional translator. The number of errors and typos is too large to be pointed out in detail here.

Abstract:

The abstract is very hard to understand due to many flows regarding the English language. Please consider revision by a native speaker before eventual publication.

Introduction:

- “is a chronic inflammatory degenerative disease. AAA is multifactorial disease with relation to both genetic and environmental risk” is a conflicting statement. I would suggest that the disease is characterized by a chronic inflammatory component with a degenerative component and that its origin is multifactorial …

- p2L41/42 “prevalence of AAA in men aged 65 and more between 1-2%” and “in cancer patients 100 times higher” -> prevalence of 100-200%? This does not make sense

- p2L48,49: the references #7,8 focus on co-prevalence and not incidence. Please rephrase. This mix between incidence and prevalence occurs a couple of times throughout the manuscript.

Patients and Methods:

- It remains unclear why the authors chose to use a HF group in addition. This needs to be explained.

- Unfortunately, it remains unclear whether patients with ruptured AAA were included or not, and if not, which I assume after reading the manuscript, why they were excluded. This also needs to be explained.

- it remains completely elusive, what types of cacners were included In the analysis and why. All types of cancers? ICD codes?

- Section 2.2 presents results acquired by the authors and should go into the results section.

Results:

- Table 1: please explain, why the matched HF group shows a significantly lower number of patients with the risk factors dyslipidemia and diabetes? Also the rate of missing data should be included for the risk factors, too.

- p5L159“A total of 372 (45.2%) AAA, 343 (41.37%) HF, and 294 (35.7%) controls were diagnosed 157

with malignant cancer (p <.001)” for what difference does the p-value stand? This is crucial information.

- p5L161: “177 (21.5%), between 6 months before and after the index date in 103 (12.5%), and 6 161

months after in 92 (11.2%)” the more common way of presenting data would be that these numbers add  up to 100% of the 372 cancer cases in the AAA group.

- From the methods section it remains unclear why the authors choose three different time points for the occurrence of cancer in relation to AAA. Why 6 months intervals? Please explain.

- table 2: a more common way to picture ORs would be a forest plot. Please consider revision. Also, Smoking and Alcohol has many missings (>50%). How was it possible to do adjusted OR with so many missings? Especially interesting with lung cancer being one of the most common ones.

- table 3 is missing a legend for a,b,c

- wouldn’t a more natural way of data analysis suggest to do a multivariate analysis on the occurrence of cancer regarding the different items investigated here? Please consider for revision.

- figure 2: have you considered looking at the cancer-free and AAA-free realted survival in the AAA cohort? This would be important information to the readership and would add to the current literature.

- generally, the authors mention many of the data presented in tables additionally in the text. Please consider shortening the redundancies.

- at 5 years follow-up, less than 25% of the patients initially included are at risk. Any comparison of survival rates does not make sense here…

Discussion:

- Unfortunately, the only lesson to be learned from the study is the high co-prevalence of cancer and AAA, data previsouls unclear. This should be emphasized.

- “Therefore, the finding from this study suggested that the coexistence of two diseases is not only occasional or colluded by the old age of patients but strongly associated.” I do not see statistical test verifying this statement. Please explain.

- Generally, in the discussion, authors refer to their presented data in figures or tables.

- AAA has a higher cancer risk before the index date. Shouldn’t it be mentioned that maybe due to cancer and certain examinations done because of it, it is more likely to detect AAA? If it is this way, cancer „causes“ AAA and not the other way around.

Author Response

Response to Reviewer 2 Comments

Thank you for your comments. We made the following changes:

The authors cover an interesting issue on regarding the co-prevalence of AAA and cancer and the possible implication for clinical decision making by investigating death rates in three different selected populations. This data is currently missing and should be of great interest to clinical practitioners. Although of general interest, the study harbors many major and minor problems in its current form.

Please find a list of topics that need to be addressed before eventual publication. PLease do consider an extensive revision of your manuscript, since the data would be interesting to the readership. 

General:

- The English language is hard to understand and covers a variety of syntax and grammatical errors as well as typos. Please revise with a native speaker or professional translator. The number of errors and typos is too large to be pointed out in detail here.

 Response: Our manuscript underwent professional English editing again as recommended.

Abstract:

The abstract is very hard to understand due to many flows regarding the English language. Please consider revision by a native speaker before eventual publication.

 Response: Our manuscript underwent professional English editing again as recommended.

Introduction:

- “is a chronic inflammatory degenerative disease. AAA is multifactorial disease with relation to both genetic and environmental risk” is a conflicting statement. I would suggest that the disease is characterized by a chronic inflammatory component with a degenerative component and that its origin is multifactorial …

 Response: We revised the sentence according to the reviewer’s suggestion. (P1L39-41)

- p2L41/42 “prevalence of AAA in men aged 65 and more between 1-2%” and “in cancer patients 100 times higher” -> prevalence of 100-200%? This does not make sense

Response: It seems that simple multiplication cannot be applied between the prevalence and the incidence. To make it clear, we added the annual incidence of AAA.

- p2L48,49: the references #7,8 focus on co-prevalence and not incidence. Please rephrase. This mix between incidence and prevalence occurs a couple of times throughout the manuscript.

Response: As the reviewer pointed out, reference #8 reported only the prevalence, not the incidence. We removed the reference on the incidence from the sentence. The manuscript was also checked and corrected for consistency so that prevalence and incidence were used appropriately.

Patients and Methods:

- It remains unclear why the authors chose to use a HF group in addition. This needs to be explained.

Response: We added the reason in the Methods section. We used the HF group as the control group for comparison because the association of HF with cancer is relatively well-known. (P3L118-119)

- Unfortunately, it remains unclear whether patients with ruptured AAA were included or not, and if not, which I assume after reading the manuscript, why they were excluded. This also needs to be explained.

Response: As stated in the Methods section, ruptured AAA was included by using ICD codes I713 and I718 (p3L107).

- it remains completely elusive, what types of cacners were included In the analysis and why. All types of cancers? ICD codes?

Response: As stated in the Methods section (p3L129), all types of cancers were included by using ICD-10 codes C00-97.

- Section 2.2 presents results acquired by the authors and should go into the results section.

Response: We moved section 2.2 to the Results section 3.1.

Results:

- Table 1: please explain, why the matched HF group shows a significantly lower number of patients with the risk factors dyslipidemia and diabetes? Also the rate of missing data should be included for the risk factors, too.

Response: All the risk factors could not be exactly matched to minimize the loss of the study population. We matched sex exactly, age within 5 y, and other risk factors, including hypertension, DM, dyslipidemia, smoking, and alcohol, as much as possible. We are unaware of the exact reason why the HF group had a lower number of risk factors, but we assumed that we included relatively young patients in the HF group. Smoking and alcohol history could be found only in patients who received national health examinations by the government. Missing data was in found in 37.9% of the patients. This was added in the Results section (p4L170). The rate of missing data of the three groups is described in Table 1.

- p5L159“A total of 372 (45.2%) AAA, 343 (41.37%) HF, and 294 (35.7%) controls were diagnosed 157 with malignant cancer (p <.001)” for what difference does the p-value stand? This is crucial information.

Response: Like other risk factors, the prevalence of malignant cancer in the three groups was compared, and the P-value stands for the statistical significance of the difference. The P-value was calculated via a generalized linear model (GEE method) considering paired data.

- p5L161: “177 (21.5%), between 6 months before and after the index date in 103 (12.5%), and 6 months after in 92 (11.2%)” the more common way of presenting data would be that these numbers add  up to 100% of the 372 cancer cases in the AAA group.

Response: We changed the percentage by changing the denominator in the three groups to make the sum 100%.

- From the methods section it remains unclear why the authors choose three different time points for the occurrence of cancer in relation to AAA. Why 6 months intervals? Please explain.

Response: Both AAA and cancer progress slowly, and it is not easy to define the temporal relationship of development between them. Therefore, we could only describe the temporal relationship of the detection of both diseases. Those found together during the workup of certain disease were defined as being discovered simultaneously. Cancer workup was not generally performed when AAA or HF was discovered, and cases up to 6 mo before and after were included in consideration of the time required for the workup of other diseases, including cancer.

- table 2: a more common way to picture ORs would be a forest plot. Please consider revision. Also, Smoking and Alcohol has many missings (>50%). How was it possible to do adjusted OR with so many missings? Especially interesting with lung cancer being one of the most common ones.

Response: As the reviewer suggested, ORs were depicted with a forest plot (Figure 2). Because there were so many missing values with smoking and alcohol, we could not match those risk factors when defining the study population for fear of greatly reducing the number of AAA patients. However, we thought this information was crucial to both cancer and AAA and presented an adjusted OR with smoking and alcohol in available patients.

- table 3 is missing a legend for a,b

Response: We added the legend.

- wouldn’t a more natural way of data analysis suggest to do a multivariate analysis on the occurrence of cancer regarding the different items investigated here? Please consider for revision.

Response: We defined the outcome with the development of cancer as a single dependent variable. Therefore, we believe that multivariate analysis was not indicated in this study. Instead, we matched the patients according to the risk factors and performed multilinear regression, which is presented as adjusted ORs.

- figure 2: have you considered looking at the cancer-free and AAA-free realted survival in the AAA cohort? This would be important information to the readership and would add to the current literature.

Response: Unfortunately, the National Insurance service regulation did not allow us to perform further data processing for administrative reasons. We will proceed with further studies with new contracts before trying to present the data in the near future.

- generally, the authors mention many of the data presented in tables additionally in the text. Please consider shortening the redundancies.

Response: As the reviewer suggested, we shortened the sentences presented in the tables.

- at 5 years follow-up, less than 25% of the patients initially included are at risk. Any comparison of survival rates does not make sense here…

Response: We agree that less than 25% of the patients initially included were at risk at 5 y of follow-up and that the survival comparison seems less meaningful. It is possible that the inclusion of all types of cancer with heterogeneous survival rates diluted the difference in survival. Although the relationship is unclear, we thought that it was worthwhile to present how the occurrence of cancer affects the survival of AAA patients and control groups.

Discussion:

- Unfortunately, the only lesson to be learned from the study is the high co-prevalence of cancer and AAA, data previsouls unclear. This should be emphasized.

Response: We agree with the reviewer’s point that the only thing we can be sure of is the high co-prevalence of cancer and AAA. We added and emphasized the high co-prevalence of cancer and AAA in the Discussion section (P13L363-365, limitation).

- “Therefore, the finding from this study suggested that the coexistence of two diseases is not only occasional or colluded by the old age of patients but strongly associated.” I do not see statistical test verifying this statement. Please explain.

Response: First, we matched the age and sex between the groups. Even after adjusting for the effect of age, the probability of the coexistence of two diseases was high with statistical significance. Therefore, we concluded that the coexistence of two diseases was not occasional or colluded by old age.

- Generally, in the discussion, authors refer to their presented data in figures or tables.

Response: We revised the Discussion section to refer to the figures or tables.

- AAA has a higher cancer risk before the index date. Shouldn’t it be mentioned that maybe due to cancer and certain examinations done because of it, it is more likely to detect AAA? If it is this way, cancer „causes“ AAA and not the other way around.

Response: We agree to the reviewer’s point. Cancer or AAA detection rates may increase with more imaging workup. We added it to the Discussion section. It is true that more cancers were discovered during the workup of AAA, which is suggested by Table 2. One strong point of our study is the long-term follow-up. This study population comprises of 13 years of cohorts. From a chronological point of view, it is reasonable to think that cancer discovered during the workup of AAA occurred long before the AAA diagnosis. If the incidence of cancer was not actually high in AAA patients, but simply that much more cancer cases were detected during the AAA workup process, the probability of cancer detected after the index date in the control group should be much higher; however, this was not the case. Subsequent cancer incidence per 100 person-years (Table 4) and cancer-related mortality (Table 6) was higher in the AAA group than in the control group. We believe that this can support the higher probability of the coexistence of cancer in AAA patients when comparing that in people of the same age group.

In AAA group, most of cancer was detected or diagnosed within 6 months of AAA diagnosis. Therefore, in terms of temporal relationship, cancer seems to precede AAA, though the causation remains unclear.

Round 2

Reviewer 2 Report

All my previous querries have been sufficiently adressed. Good luck with future endeavours.